# E8002 Reduces Adhesion Formation and Improves Joint Mobility in a Rat Model of Knee Arthrofibrosis

**DOI:** 10.3390/ijms23031239

**Published:** 2022-01-22

**Authors:** Seiya Takada, Kentaro Setoyama, Kosuke Norimatsu, Shotaro Otsuka, Kazuki Nakanishi, Akira Tani, Tomomi Nakakogawa, Ryoma Matsuzaki, Teruki Matsuoka, Harutoshi Sakakima, Salunya Tancharoen, Ikuro Maruyama, Eiichiro Tanaka, Kiyoshi Kikuchi, Hisaaki Uchikado

**Affiliations:** 1Department of Systems Biology in Thromboregulation, Kagoshima University Graduate School of Medical and Dental Sciences, 8-35-1 Sakuragaoka, Kagoshima 890-8520, Japan; karaagetantou0110@gmail.com (S.T.); k3360022@kadai.jp (S.O.); maruyama-i@eva.hi-ho.ne.jp (I.M.); 2Division of Laboratory Animal Science, Natural Science Center for Research and Education, Kagoshima University, 8-35-1 Sakuragaoka, Kagoshima 890-8520, Japan; seto@m.kufm.kagoshima-u.ac.jp; 3Course of Physical Therapy, School of Health Sciences, Faculty of Medicine, Kagoshima University, 8-35-1 Sakuragaoka, Kagoshima 890-8544, Japan; k6745961@kadai.jp (K.N.); k9378361@kadai.jp (K.N.); k3694885@kadai.jp (A.T.); 1033.msr@gmail.com (T.N.); ryoma.m1026@outlook.jp (R.M.); k3988526@kadai.jp (T.M.); sakaki@health.nop.kagoshima-u.ac.jp (H.S.); 4Department of Pharmacology, Faculty of Dentistry, Mahidol University, Bangkok 10400, Thailand; salunya.tan@mahidol.edu; 5Division of Brain Science, Department of Physiology, Kurume University School of Medicine, 67 Asahi-machi, Kurume, Fukuoka 830-0011, Japan; eacht@med.kurume-u.ac.jp; 6Department of Neurosurgery, Kurume University School of Medicine, 67 Asahi-machi, Kurume, Fukuoka 830-0011, Japan; 7Uchikado Neuro-Spine Clinic, 1-2-3 Naka, Hakata-ku, Fukuoka 812-0893, Japan

**Keywords:** anti-adhesive membrane, knee adhesion, E8002, arthrofibrosis, myofibroblast

## Abstract

Knee arthrofibrosis is a common complication of knee surgery, caused by excessive scar tissue, which results in functional disability. However, no curative treatment has been established. E8002 is an anti-adhesion material that contains L-ascorbic acid, an antioxidant. We aimed to evaluate the efficacy of E8002 for the prevention of knee arthrofibrosis in a rat model, comprising injury to the surface of the femur and quadriceps muscle 1 cm proximal to the patella. Sixteen male, 8-week-old Sprague Dawley rats were studied: in the Adhesion group, haemorrhagic injury was induced to the quadriceps and bone, and in the E8002 group, an adhesion-preventing film was implanted between the quadriceps and femur after injury. Six weeks following injury, the restriction of knee flexion owing to fibrotic scarring had not worsened in the E8002 group but had worsened in the Adhesion group. The area of fibrotic scarring was smaller in the E8002 group than in the Adhesion group (*p* < 0.05). In addition, the numbers of fibroblasts (*p* < 0.05) and myofibroblasts (*p* < 0.01) in the fibrotic scar were lower in the E8002 group. Thus, E8002 reduces myofibroblast proliferation and fibrotic scar formation and improves the range of motion of the joint in a model of knee injury.

## 1. Introduction

Arthrofibrosis is a disease characterized by pain, loss of range of motion (ROM), warmth, exudation, swelling, and the development of fibrous scarring [1]. The postoperative incidence of arthrofibrosis has been reported to be 1–13% after total knee arthroplasty, 0–4% after ligament injury, such as to the anterior cruciate ligament, and 7% after high-energy knee fracture [2,3,4]. The formation of excessive scar tissue in the knee joint leads to a loss of ROM, tissue contracture, greater pain, and impairments in activities of daily living, such as standing up, walking, and climbing stairs [5,6,7,8]. In consequence, knee joint fibrosis represents a serious challenge for orthopaedic and rehabilitation departments worldwide.

Although the aetiology remains to be fully elucidated, one possible cause of arthrofibrosis is that tissue damage stimulates immune cells, inducing oxidative stress and the production of pro-inflammatory cytokines, such as platelet-derived growth factor (PDGF), transforming growth factor-beta (TGF-beta), and interleukin (IL)-1 [9,10]. This causes fibroblasts to differentiate into myofibroblasts, which produce excessive amounts of extracellular matrix (ECM), including collagen and elastin, which limits joint capsule flexibility and joint movement [9,10]. Although ECM is required for healing and wound repair, dysregulation of its production and degradation leads to pathological fibrosis [10,11]. The approaches to the treatment of knee fibrosis include manipulation under anaesthesia, debridement, and physical therapy [12], but the ideal measure would be to prevent the development of arthrofibrosis. Hydrogel microspheres, temperature-sensitive anti-adhesive poloxamer (TAP) hydrogel, hyaluronic acid, carboxymethylcellulose, decorin, chitosan, lovastatin, rapamycin, and hydroxycaptotensin have been shown to have anti-adhesive effects [13,14,15,16,17,18,19,20], and chitosan and TAP hydrogel have been shown to be effective in clinical trials [14,17]. However, these agents have yet to be commercialized, and no surgical means of preventing adhesions following knee surgery have been established.

E8002 is a three-layered anti-adhesion material that was previously known as nDM-14R [21]. Its central layer consists of pullulan, which is harmless, easily absorbed, and used in foods and pharmaceuticals [22], and the surface layers comprise l-lactide, glycolide, and ε-caprolactone copolymer, which is produced by ring-opening polymerization (Taki Chemical, Kakogawa, Japan), catalysed by tin octoate [Sn(O_2_C_8_H_15_)_2_] [22]. The material comprising the central layer dissolves quickly under moist conditions, while that used for the surface layers dissolves slowly [22]. In addition, both the central and surface layers contain L-ascorbic acid [22]. Previously, we reported that E8002 inhibits peripheral nerve adhesion in a model of sciatic nerve injury and epidural scar formation after vertebroplasty in rats [22,23]. E8002 has been shown to act as a slowly dissolving temporary physical barrier that prevents the formation of inter-tissue adhesions, and to inhibit fibrous scar formation via the tissue plasminogen activator-mediated fibrinolytic action of L-ascorbic acid [22,23]. However, there have been no studies of the use of E8002 for the prevention of fibrosis in the knee joint. Therefore, in the present study, we evaluated the effects of E8002 in a rat model of knee joint fibrosis.

## 2. Results

### 2.1. E8002 Treatment Reduces the Post-Injury Limitation of Joint Flexion

To determine whether E8002 maintains postoperative knee mobility, we examined the range of motion of the stifles of rats over time. The rats were randomly allocated to an E8002 group (*n* = 8) or Adhesion group (*n* = 8) and the uninjured hindlimbs of eight randomly selected rats from the two groups were regarded as No Adhesion controls (*n* = 4 from each group). The acute angle formed by the femoral greater trochanter, femoral lateral epicondyle, and fibular external capsule was measured. The flexion restriction angle of the stifle in the No Adhesion group did not change significantly over the 6 weeks of the study (pre-injury: 38.33 ± 1.26°, 6 weeks: 39.10 ± 1.06°). The Adhesion group showed an increase in knee flexion restriction angle with time following injury (pre-injury: 37.31 ± 1.87°, 6 weeks: 45.52 ± 1.14°), whereas the E8002 group did not show an increase, despite the injury (pre-injury: 37.04 ± 1.72°, 6 weeks: 38.16 ± 0.87°) (Figure 1A,B). The results of two-way repeated-measures analysis of variance were *p* = 0.056 for the effect of group, *p* = 0.001 for the effect of time, and *p* < 0.001 for the interaction. Specifically, the knee flexion restriction angle of the E8002 group was lower than that of the Adhesion group after 6 weeks, and was comparable to that of the No Adhesion group (Tukey’s range test: Adhesion group vs. No Adhesion group: *p* < 0.001, Adhesion group vs. E8002 group: *p* < 0.001, No Adhesion group vs. E8002 group: *p* = 0.643) (Figure 1B). Thus, the insertion of E8002 at the site of linear scarring following knee arthroplasty can reduce subsequent restriction of knee joint motion.

### 2.2. E8002 Inhibits Myofibroblast Proliferation and Scar Tissue Formation

The cause of the E8002-associated reduction in knee flexion restriction angle was then investigated morphologically and histologically. Sagittal sections of the normal and injured limbs obtained 6 weeks after injury are shown in Figure 2A. The stifle joint epiphyses were normal in the No Adhesion group, whereas connective tissue proliferation was present in the Adhesion and E88002 groups. However, the area of scar tissue area in the E8002 group was significantly smaller than that in the Adhesion group (Adhesion group: 2.73 ± 0.34 mm^2^, E8002 group: 1.41 ± 0.23 mm^2^, Student’s *t*-test: *p* < 0.01) (Figure 2B).

Haematoxylin and eosin (HE)-stained images of the fibrous scars are shown in Figure 3A. Because no scarring developed in the No Adhesion group, we counted the numbers of haematoxylin-stained nuclei in the fibrous scars of the other two groups, which were considered to be the nuclei of fibroblasts [20]. There were significantly fewer nuclei in the E8002 group than in the Adhesion group (Adhesion group: 5495 ± 244 cells, E8002 group: 4540 ± 349 cells, Student’s *t*-test: *p* = 0.04) (Figure 3A,B).

Immunostaining for α-smooth muscle actin (α-SMA) was next performed to show the presence of myofibroblasts in the scar tissue (Figure 3C). Myofibroblasts are a key mediator of excessive fibrotic scar formation [24,25]. The number of myofibroblasts was significantly lower in the E8002 group then in the Adhesion group (Adhesion group: 1344 ± 186 cells, E8002 group: 601 ± 182 cells, Mann–Whitney U test: *p* = 0.02) (Figure 3C,D). These data suggest that E8002 reduces excessive scar tissue formation by reducing fibroblast and myofibroblast proliferation.

## 3. Discussion

E8002 is a novel multilayer membrane that has been shown to have anti-adhesive effects in rat models of appendage adhesion, vertebral arch resection, and sciatic nerve adhesion [21,22,23]. In the present study, we have confirmed the efficacy of E8002 using a rat model of arthrofibrosis. Specifically, we have shown that E8002 reduces the limitation to the ROM of the stifle induced by surgical injury and that this may be mediated through decreases in the number of myofibroblasts and the amount of scar tissue.

Arthrofibrosis is a fibrotic joint disorder that begins as an inflammatory response to an insult, such as injury, surgery, or infection [9]. The treatments for arthrofibrosis include manipulation under anaesthesia and debridement, but these procedures may be associated with complications such as neurovascular disorders [2,3,4]. In addition, excessive physical therapy may trigger an inflammatory response and thereby worsen the arthrofibrosis [12,26,27]. Therefore, the implantation of E8002 may prevent the development of postoperative arthrofibrosis and facilitate ‘moderate’ physical therapy after treatment.

Myofibroblasts are important for wound healing but are usually absent in healthy tissues [28]. They are the primary effector cells of fibrosis, which involves the deposition of excess ECM and dense fibrous collagen [11,29,30]. Reactive oxygen species (ROS) and pro-inflammatory cytokines, such as TGF-β, IL-1β, and IL-6, play important roles in myofibroblast proliferation [28]. However, E8002 contains ascorbic acid, which is a potent antioxidant [31] and prevents ROS-induced oxidative damage to biological macromolecules, such as DNA, lipids, and proteins [32]. Ascorbic acid has been shown to cause the proliferation of myofibroblasts in collaboration with TGF-β in vitro [33], but the model used in this study did not mimic the prevalent microenvironment associated with tissue injury. However, in a double-blind, placebo-controlled, randomized trial, oral ascorbic acid was found to significantly reduce the incidence of knee arthrofibrosis 1 year following injury [34]. Moreover, the consumption of vitamin C has been reported to reduce complex regional pain syndrome after total knee arthroplasty [35]. Finally, we have previously shown that E8002 inhibits peripheral nerve adhesion by increasing the fibrinolytic effects of ascorbic acid [23]. The present findings are consistent with the fibrinolytic effect of E8002 [23] and suggest that E8002 may inhibit myofibroblast proliferation by reducing postoperative oxidative stress, rather than directly inhibiting myofibroblast proliferation, through the maintenance of an effective local concentration of ascorbic acid.

Positive effects of hydrogel microspheres, TAP hydrogel, hyaluronic acid and carboxymethylcellulose, decorin, chitosan, lovastatin, rapamycin, and hydroxycaptotensin have been identified in animals, but most of these have yet to be tested in human patients [13,14,15,16,17,18,19,20], and to the best of our knowledge, there have no studies of the use of these agents in clinical practice for the prevention of arthrofibrosis following knee joint surgery in humans. The reasons for the delay in the trial of these substances in clinical practice is unknown, but this serves to emphasize the necessity for the development of new methods for the control of arthrofibrosis. As has been frequently demonstrated previously, imperfect knee function is often a source of stress and may reduce the quality of life of patients [5,6,7,8]. As treatment technologies develop further, the level of joint function required by patients would be expected to increase, and therefore the control of arthrofibrosis using biological or synthetic materials and drugs will be of increasing importance. Therefore, long-term studies of knee joint mobility will be necessary in the future.

There were several limitations to the present study. First, we did not characterize the changes in the tissues over time or how E8002 affects the surrounding tissues. This is because the number of animals per group was restricted for ethical reasons. Second, the importance of oxidative stress in models of knee joint injury requires further study. However, it is not easy to directly assess oxidative stress caused by ROS or free radicals because ROS are generated immediately after injury and are likely to be undetectable after 6 weeks. Third, the anatomy and mechanics of the rat stifle differ to those of the human knee: rats are quadrupedal, and therefore, their forelimbs are also weight-bearing. Fourth, arthrofibrosis can have a number of different aetiologies, whereas the present model reflects only one of these. Finally, we have not evaluated the effects of the physical properties of E8002 on knee mechanics. There was a slight increase in knee flexion restriction in the E8002 group 2 weeks after the procedure; therefore, it is possible that it may have a small effect, but its softness and thinness would limit this, and because it gradually disappears from the joint, any effect is likely to be transient.

In conclusion, the results of the present study suggest that E8002 reduces myofibroblast proliferation and fibrotic scar formation and maintains the joint ROM in a rat model of knee injury. These findings suggest that a combination of physical and pharmacological means of reducing the postoperative formation of adhesions and scarring may be effective.

## 4. Materials and Methods

### 4.1. Animals and Experimental Groups

We studied 16 male, 8-week-old Sprague Dawley rats (250–300 g) that were purchased from Charles River Laboratories (Yokohama, Kanagawa, Japan). The rats were housed under a 12-h light/dark cycle and controlled temperature (22.0 ± 1.0 °C), with free access to food and water. They were randomly allocated to either an E8002 group (*n* = 8) or an Adhesion group (control group: *n* = 8). The healthy hindlimbs of eight randomly selected rats were classified as the No Adhesion group (*n* = 4, E8002; *n* = 4, Adhesion). The experimental protocol was approved by the Kagoshima University Animal Ethics Committee (approval number MD18025; date: 25 June 2018).

### 4.2. Knee Arthrofibrosis Model

A knee arthrofibrosis model was created, as described previously [36]. Briefly, rats were anesthetized with 1.5%–2.0% isoflurane (Pfizer Inc., Shibuya-ku, Tokyo, Japan) using an MK-A110 Small Animal Anesthetizer (Muromachi Kikai Co., Ltd., Chuo-ku, Tokyo, Japan). The left stifle of each was shaved using electric clippers, sterilized with 70% ethanol, and draped under aseptic conditions. After the skin was incised, the stifle was opened using a medial parapatellar approach and the medial and lateral sides of the femoral condyle were exposed. Three cuts were then made using a scalpel on the articular surface of the femur and quadriceps muscle 1 cm above the patella, causing bleeding. In the E8002 group, a rectangle of E8002 was inserted (width: 5 mm, height: 15 mm) between the femur and quadriceps (Figure 4), but in the Adhesion group, nothing was inserted. The muscle and skin of all the rats were then sutured using Monocryl 4-0 (Ethicon, Somerville, NJ, USA). The left leg was not fixed, so that it could be moved freely. Six weeks after the procedure, none of the rats showed evidence of wound infection.

### 4.3. Joint Angle Measurement

The flexion angle of the stifle of each rat was measured every 2 weeks from prior to injury to 6 weeks afterwards under anaesthesia using 1.5%–2.0% isoflurane. The trunk and left leg were fixed, and a force of 0.5 N was applied to the distal tibia using a Wds-180A Compact Digital Indicator (Kyowa Electronic Instruments Co., Ltd., Chofu City, Tokyo, Japan). It was thought that the application of too much pressure would elongate the soft tissues around the joint and affect the ROM of the joint. Preliminary measurements were made at 1.0 N, 0.5 N, and 0.3 N, and 0.5 N. At 0.3 N, the reaction force was weak, and it was thought to be from the fatty tissue and muscles on the posterior surface of the thigh, not from the fibrous scar. 1N applied strong extension stress to the fibrous scar, and it was thought to be a range-of-motion expansion training. 0.5 N felt moderate resistance. Therefore, 0.5N was adopted. A similar force was used in a recent study [37]. The acute angle between the long axis of the femur and the tibia was measured, using the femoral greater trochanter, femoral lateral epicondyle, and external capsule as landmarks. Two investigators measured the angles formed by these three points using Scion Image Software 4.0.3 (Scion Corp, Frederick, MD, USA) and the mean value was calculated.

### 4.4. Histological and Immunohistochemical Evaluation

After the final measurement of the joint angle, the stifle was dissected from the mid-point to the distal patella under terminal anaesthesia. The samples were immersed overnight in 4% paraformaldehyde in 0.1 M phosphate buffer (pH 7.4) at 4 °C, then decalcified using Kalchitox (hydrochloric acid 4.8%, disodium EDTA 0.1% aqueous solution; Fujifilm Wako Pure Chemical Industries, Osaka City, Osaka, Japan) for 72 h, and neutralized using 5% sodium sulphate solution for 24 h. They were then cut as symmetrically as possible along the long axis of the femur and paraffin-embedded. Blocks containing the medial femoral condyles were then cut into 4-µm sections ~100–150 µm medial to the mid-femur, using the patellar tendon transition as a landmark. Aldehyde-fuchsin-Masson-Goldner staining and HE staining were performed. The scar tissue on the aldehyde-fuchsin-Masson-Goldner-stained sections was magnified 4× and photographed using an Olympus DP21 microscope (Shinjuku-ku, Tokyo, Japan).

Sections were also immunostained using rabbit anti-α-SMA antibody to identify myofibroblasts (Cosmo Bio Co., Ltd. Koto-ku, Tokyo, Japan). Following deparaffinization and rehydration, the endogenous peroxidase activity was blocked by incubation in methanol containing 3.0% hydrogen peroxide for 10 min. The sections were then rinsed three times (5 min each) with PBS (pH 7.6) and blocked with 10% skimmed milk in PBS for 20 min. The sections were individually incubated at 4 °C overnight in the α-SMA antibody (1:200), washed in PBS three further times, and then incubated for 60 min with goat anti-rabbit IgG conjugated to a peroxidase-labelled dextran polymer (Dako EnVision+ System-HRP Labelled Polymer Anti-Rabbit; Agilent Technologies, Santa Clara, CA, USA), according to the manufacturer’s instructions. Finally, the sections were rinsed with PBS and their immunoreactivity was visualized by diaminobenzidine staining.

### 4.5. Quantitative Analysis

The area of the scar tissue was analysed using Scion Image Software 4.0.3. The fibroblasts and myofibroblasts were photographed at three random locations using the DP21 microscope at 40× magnification. Each type of cell was counted, and the mean numbers were calculated.

### 4.6. Statistical Analysis

All data were subjected to the Shapiro–Wilk test, then two-way repeated-measures analysis of variance was used to determine the effect of E8002 on the change in knee flexion over time. Tukey’s range test was used for post hoc testing. Student’s *t*-test was used to compare the area of scarring and the number of fibroblasts between the E8002 and Adhesion groups, and the Mann–Whitney U test was used to compare the number of myofibroblasts. Two-sided *p*-values < 0.05 were considered to represent statistical significance. Data are expressed as mean ± standard error. The data were analysed using GraphPad prism 9.1.0 (221) (Graphpad Holdings, LLC, San Diego, CA, USA).

## Figures and Tables

**Figure 1 ijms-23-01239-f001:**
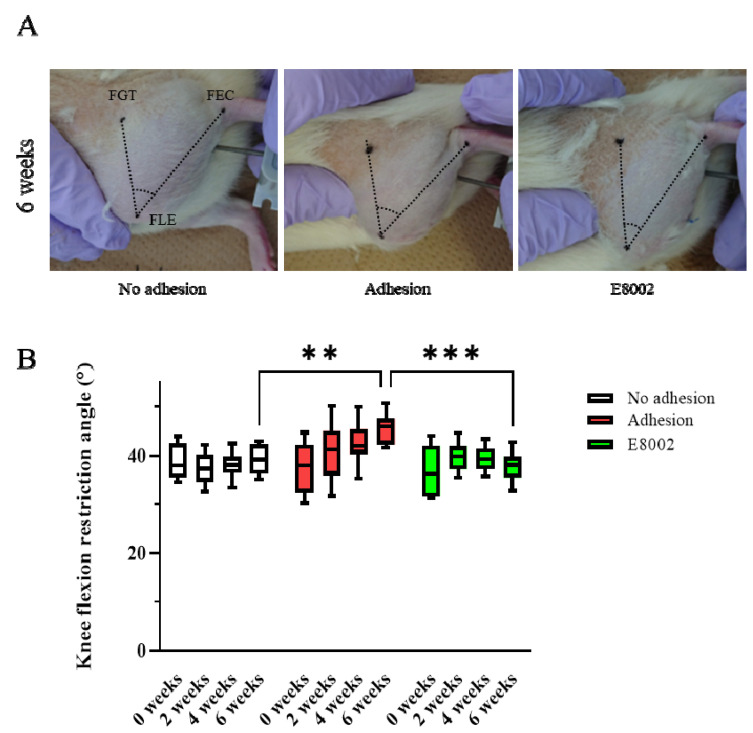
E8002 preserves joint mobility after stifle injury. (**A**) Restriction of joint flexion after stifle injury was compared between the No Adhesion, Adhesion, and E8002 groups over 6 weeks (n = 8 per group). The acute angle formed by the three points of the femoral greater trochanter, femoral lateral condyle, and tibial epicondyle was measured. (**B**) Knee flexion in each group at each time point. Data are mean ± SE. The results of two-way RM-ANOVA were p = 0.056 for the effect of group, *p* = 0.001 for the effect of time, and *p* < 0.001 for the interaction. Post hoc testing (Tukey’s range test) indicated a significantly greater restriction of knee flexion at 6 weeks in the Adhesion group (Adhesion group vs. No Adhesion group: *p* < 0.001, Adhesion group vs. E8002 group: *p* < 0.001, No Adhesion group vs. E8002 group: p = 0.643). ** *p* < 0.05, *** *p* < 0.01.

**Figure 2 ijms-23-01239-f002:**
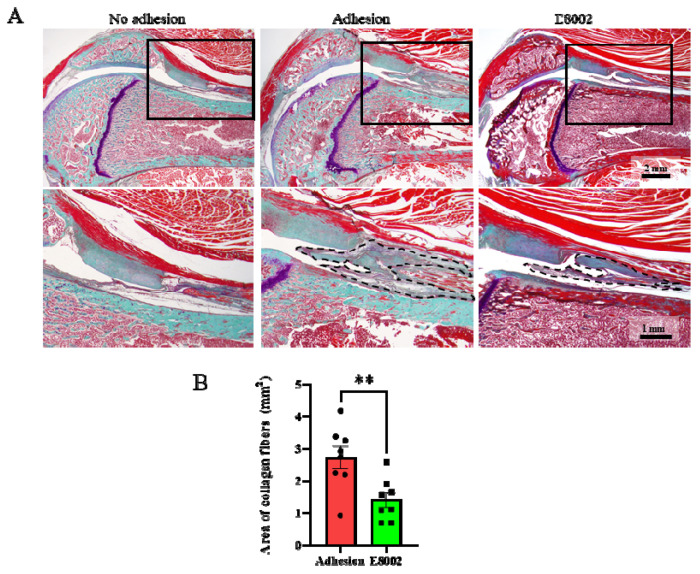
E8002 reduces the formation of fibrous scars after stifle injury. (**A**) Upper row: representative photographs of fibrous scars stained using aldehyde-fuchsin-Masson-Goldner stain. Lower row: higher magnification of the areas indicated by solid black boxes on the photomicrographs above. Black dotted lines outline the areas of fibrotic scarring. (**B**) E8002 significantly reduced the formation of fibrotic scars at the 6-week time point. Data are mean ± SE. ** *p* < 0.01 (Student’s *t*-test).

**Figure 3 ijms-23-01239-f003:**
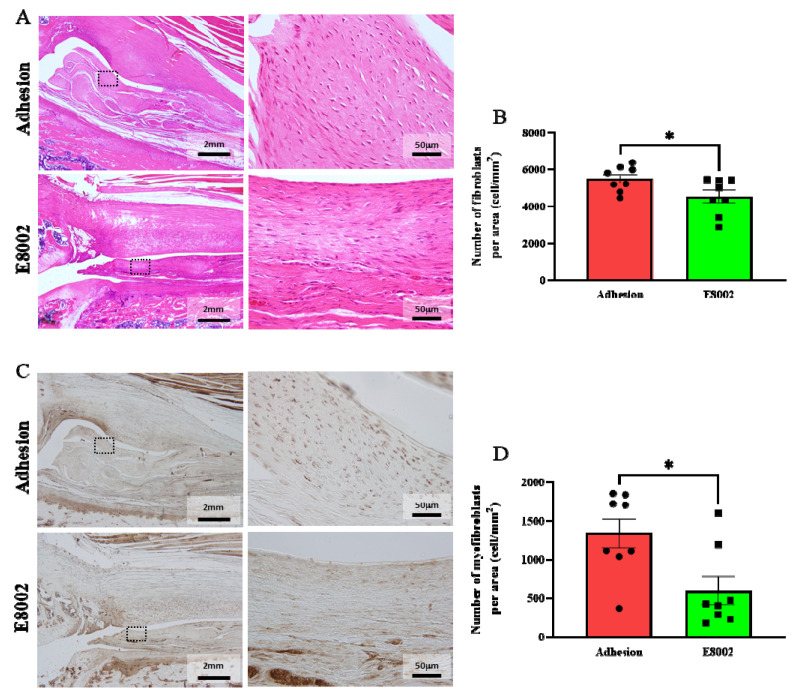
E8002 inhibits the proliferation of myofibroblasts. (**A**) Left: representative photomicrographs of fibroblasts in fibrotic scars (haematoxylin and eosin staining). Right: magnified images of the areas within the black dotted frames. (**B**) Comparison of the mean number of fibroblasts at three different locations. E8002 reduced the number of fibroblasts in the fibrotic scars 6 weeks after injury. (**C**) Left: representative photomicrographs of myofibroblasts in fibrotic scars (α-smooth muscle actin immunostaining. Right: magnified images of the areas within the black dotted frames. (**D**) Comparison of the mean number of myofibroblasts at three different locations. E8002 reduced the number of fibroblasts in the fibrotic scars 6 weeks after injury. Data are mean ± SE. * *p* < 0.05 (Figure 3B: Student’s *t*-test; Figure 3D: Mann–Whitney U test).

**Figure 4 ijms-23-01239-f004:**
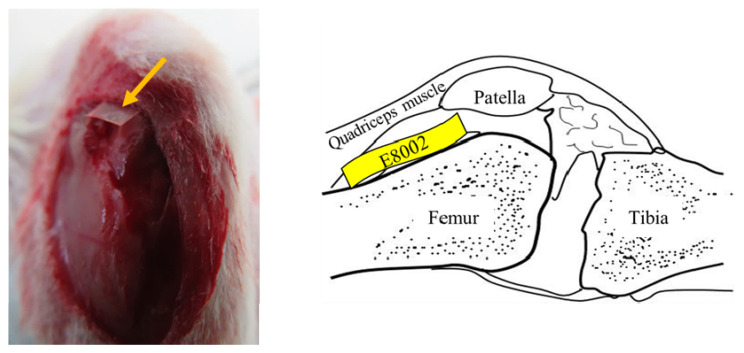
Site of insertion of the E8002 rectangles. The skin was incised to expose the surface of the stifle joint. The yellow arrows indicate the anti-adhesion E8002 membrane, which was inserted between the femur and the quadriceps muscle. The illustration on the right shows the site of insertion of the E8002 in the sagittal plane.

## Data Availability

The data that support the findings of this study are available from the corresponding author (Kiyoshi Kikuchi) upon reasonable request.

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
