# Peer review of "E8002 Reduces Adhesion Formation and Improves Joint Mobility in a Rat Model of Knee Arthrofibrosis"

_ijms, 2022, doi:10.3390/ijms23031239_

Round 1

Reviewer 1 Report

The problem of arthrofibrosis is well known to any orthopedic practitioner. The presented work is an interesting and innovative view on this issue. A relatively high percentage of complications in the form of arthrofibrosis after surgery on large joints results from the biological reaction of the patient, which is difficult to predict and modify. The introduction of implants that could modify this reaction could be of great importance.

The reviewed paper appears to be one of the series on Novel Anti-Adhesive Membrane, E8002. There are some slight and insignificant convergences between these works, especially in the introduction section. Due to the value of the presented work, I propose to consider their slight redrafting.

Author Response

RESPONSES TO REVIEWER 1

We are very grateful for the reviewer’s input. Here are the responses to their comments:

General comment: The problem of arthrofibrosis is well known to any orthopedic practitioner. The presented work is an interesting and innovative view on this issue. A relatively high percentage of complications in the form of arthrofibrosis after surgery on large joints results from the biological reaction of the patient, which is difficult to predict and modify. The introduction of implants that could modify this reaction could be of great importance.

Response: We thank the reviewer for their comments. We have addressed the specific points below.

Comment: The reviewed paper appears to be one of the series on Novel Anti-Adhesive Membrane, E8002. There are some slight and insignificant convergences between these works, especially in the introduction section. Due to the value of the presented work, I propose to consider their slight redrafting.

Response: We agree with the reviewers and have partially revised the introductory paragraphs 2, 3 and 4, lines 61-83.

Reviewer 2 Report

The authors have carried out an interesting study that may contribute to improving the treatment of a common pathology that is difficult to manage. My most sincere congratulations.

Regarding the presentation of the manuscript, it is necessary that the methods section is presented before the results, to facilitate the reading of the manuscript.

Author Response

RESPONSES TO REVIEWER 2

We are very grateful for the reviewer’s input. Here are the responses to their comments:

General comment: The authors have carried out an interesting study that may contribute to improving the treatment of a common pathology that is difficult to manage. My most sincere congratulations.

Response: We thank the reviewer for their comments. We have addressed the specific points below.

Comment: Regarding the presentation of the manuscript, it is necessary that the methods section is presented before the results, to facilitate the reading of the manuscript.

Response: We agree with the reviewer's comment and moved the method position to lines 84-170.

This manuscript is a resubmission of an earlier submission. The following is a list of the peer review reports and author responses from that submission.

Round 1

Reviewer 1 Report

Dear Authors,

Congratulations for your work. I have few comments to be addressed:

Rats were housed in cages with free access to standard materials and water --> Food, not materials

The results are poorly elaborated and details are laking

The scientific level of the English language should be improved

Thank you and best regards

Reviewer 2 Report

Takada and colleagues assess the impact of E8002 arthrofibrosis in a rat model. While parts of the study show promising results in the use of E8002 for the prevention of arthrofibrosis, there are several issues with the studies that need attention by the authors. Besides the major points outlined below, the study also lacking depth and additional approaches should be undertaken to assess the effect of E8002 on arthrofibrosis in this rat model.

  1. The authors employed extremely young rats for their studies (4 weeks old, juvenile). The use of these extremely young rats is questionable as rats are in the actively growing phase (juveniles) and clinical relevance may not be supported as arthrofibrosis is mostly observed in adult patients (18+). This is especially concerning as the original model paper (Ref 38) used adult rats for the studies, raising issues related to the authentication of the present model.
  2. As stated in the methods, the authors mentioned their concerns is it relates to the joint angle measurements in vivo (too much force could elongate the tissue). The authors state that 0.5 N was optimal in preliminary studies. Showing these preliminary data is of utmost importance to ensure that the testing device is not affecting results over the course of the experiment.
  3. Does E8002 alter the joint angle measurements? Did the authors test whether the actual physical nature of the membrane alters the knee mechanics and not necessarily the fibrotic build-up.
  4. How did the authors ensure consistent location of histologic sections? Also, why was this particular area chosen or analysis?
  5. The authors mentioned that fibroblast were counted (Fig 3). There is a lack of methods that describes how fibroblasts were identified and counted. Also, is it really about the number of fibroblasts or should the authors be concerned about the number of myofibroblasts?
  6. The authors make several statements including “E8002 is an anti-adhesion membrane containing L-ascorbic acid, an antioxidant” and “E8002 functions as a physical barrier to prevent adhesion between tissues and as an antioxidant related to its ascorbic acid content.” Some of these statements may suggest that the effects could be due to ascorbic acid. Now, due to lack of studies related ascorbic acid, the only thing that can be concluded are the physical barrier effects by E8002. So, the authors should be careful of how their conclusions are based and avoid making conclusions related to the effects of ascorbic acid.